# Quantum Calibration of Photon-Number-Resolving Detectors Based on Multi-Pixel Photon Counters

**Yujie Cai**, **Yu Chen \***, **Xiuliang Chen, Jianhui Ma, Guangjian Xu, Yujing Wu, Aini Xu** and **E Wu \***

State Key Laboratory of Precision Spectroscopy, East China Normal University, Shanghai 200062, China
* Correspondence: leon06220308@163.com (Y.C.); ewu@phy.ecnu.edu.cn (E.W.);
  Tel.: +86-18221068229 (Y.C.); +86-13764139892 (E.W.)

**Abstract:** In this paper, we reconstructed the positive operator-valued measure (POVM) of a photon-number-resolving detector (PNRD) based on a multi-pixel photon counter (MPPC) by means of quantum detector tomography (QDT) at 791 nm and 523 nm, respectively. MPPC is a kind of spatial-multiplexing PNRD with a silicon avalanche photodiode (Si-APD) array as the photon receiver. Experimentally, the quantum characteristics of MPPC were calibrated at 2 MHz at two different wavelengths. The POVM elements were given by QDT. The fidelity of the reconstructed POVM elements is higher than 99.96%, which testifies that the QDT is reliable to calibrate MPPC at different wavelengths. With QDT and associated Wigner functions, the quantum properties of MPPC can be calibrated more directly and accurately in contrast with those conventional methods of modeling detectors.

**Keywords:** photon-number-resolving detection; quantum detector tomography; multi-pixel photon counter; positive operator-valued measure

## 1. Introduction

Accurate representation of multi-photon states is one of the key tasks of modern quantum optics [1]. To represent multi-photon states, photon-number-resolving detectors (PNRDs) are widely applied in many fields [2], including the demonstration of basic principles of quantum mechanics [3], quantum computation with linear optical components [4], quantum key distribution (QKD) [5], quantum random number generation [6,7] and so forth. PNRDs can also be used for photon-counting laser ranging and three-dimensional laser imaging in long-distance, and laser radar based on coherent laser, etc. [8–12] in the field of aerospace to improve the signal to noise ratio. In addition, PNRDs also play an important role in the field of biology and medicine, such as fluorescence detection [13,14], pollutant monitoring [15], flow cytometry [16], positron emission computed tomography (PET) [17–19].

Traditional single-photon detectors such as Si-APD and InGaAs APD are unable to resolve the number of incident photons, though they are fully developed and applied in various fields [20]. In order to obtain the information of the incident photon statistics, considerable progress has been made in the development of PNRDs, including direct PNRDs [21] and multiplexed PNRDs [22]. For example, superconducting transition-edge sensor (TES) is a type of direct PNRD with high detection efficiency [23] and direct photon-number-resolving capability [24]. However, the extremely complex structure and the ultra-low working temperature limit the applications of TES. The time-multiplexed detector (TMD) [25,26] and spatial-multiplexed detector [27–29] are another type of PNRDs based on the multiplexing technique. Compared with the superconducting single-photon detector, the spatial-multiplexed single-photon detector has a relatively simpler and more compact structure, lower cost, higher stability when operated at room temperature, and a wider range of practical applications. Compared with the TMD, the spatial-multiplexed detector can conduct stable detection at a higher

repetition rate [30]. In the visible regime, the multi-pixel photon counter (MPPC) is usually used as a convenient PNRD based on spatial multiplexing [31]. MPPC has many advantages such as excellent photon-number-resolving capability, great response speed, high time resolution, and wide spectral response range. The compactness of MPPC enables it to be packaged into different devices for different demands and be applied to many research fields [32]. MPPC can also be combined with frequency up-conversion systems to achieve the infrared PNRD [6,33,34].

To characterize the detection performance of MPPCs, the traditional parameters include detection efficiency, dark count rate, after pulsing probability, crosstalk probability, etc. Among them, the detection efficiency is determined by the materials and the coating on the detector. And at different wavelengths, the detection efficiency varies. In the past, partial calibrations or elaborate models of a detector were given based on these traditional parameters, which can no longer meet the requirements of describing its quantum characteristics completely and accurately. Since the quantum detector is employed in more and more complex situations, Lundeen et al. first realized the tomography of quantum detectors in 2009 [35]. PNRD was first fully characterized by the so-called quantum detector tomography (QDT). In quantum mechanics, POVM elements can be used to describe quantum detectors. In previous work [36], the quantum state tomography (QST) has been demonstrated with MPPC, in which the quantum property of the detector was considered as a known fact. The quantum state of the incident photons was reconstructed according to the QST, but in QDT, the quantum property of the detector is unknown and the incident photon states are well-known. In 2013, the time-multiplexed superconducting nanowire single photon detector has already been well characterized by QDT at 1550 nm [37]. Comparing with the previous model description methods [38], we can characterize the quantum properties of MPPC based on spatial-multiplexing technique precisely and comprehensively through the reconstructed POVM elements at two different wavelengths. It will facilitate the application of precisely calibrated MPPC in the field of quantum information. Under very few preconditions, QDT can realize the "black box" calibration without model construction and calibration of specific parameters such as detection efficiency. The characterized detectors could also be used to prepare various non-Gaussian states [26].

Here, we characterized the photon-number-resolving capability of MPPC by reconstructing the POVM elements using the method of QDT. We obtained its quantum features at the wavelengths of 791 nm and 523 nm, respectively. Quantum calibration of this kind of detector at different wavelengths provides an experimental and theoretical basis for the applications of MPPC in quantum optics and quantum sensing.

## 2. Materials and Methods

### 2.1. Multi-Pixel Photon Counter (MPPC)

An MPPC, also known as a silicon photomultiplier (SiPM), consists of multiple avalanche photodiode (APD) pixels, as shown in Figure 1a. The effective photosensitive area of MPPC (C13366-1350GA Hamamatsu, Hamamatsu City, Japan) is $1.3 \times 1.3$ mm$^2$, and the pixel pitch is 50 μm. The number of pixels is 667 in total. Each pixel operated in Geiger mode independently for detecting the incident photons. The voltage applied on each APD is 54.1 V. And the breakdown voltage is about $51.1 \pm 5$ V. MPPC was passively quenched in the experiment when photons were detected. The deadtime of MPPC is about 70.8 ns. When the photons arrive on different Si-APD pixels within the same short time window less than 1 ns, avalanche pulses with almost the same amplitude are induced by photons with different pixels. Since the cathode and anode of all pixels are the same, MPPC would produce an avalanche pulse whose amplitude was the sum of the response from all the APD pixels. The peak amplitude voltage from MPPC is proportional to the number of detected photons, providing the photon-number-resolving capability as shown in Figure 1b. As a result, the peak voltage amplitude can directly show the information of the incident photon number. We used digital oscilloscope to capture the peak voltage amplitude of the signal pulses. The oscilloscope was

triggered by the synchronization signal from the laser source. The MPPC's output pulse duration was about 40 ns.

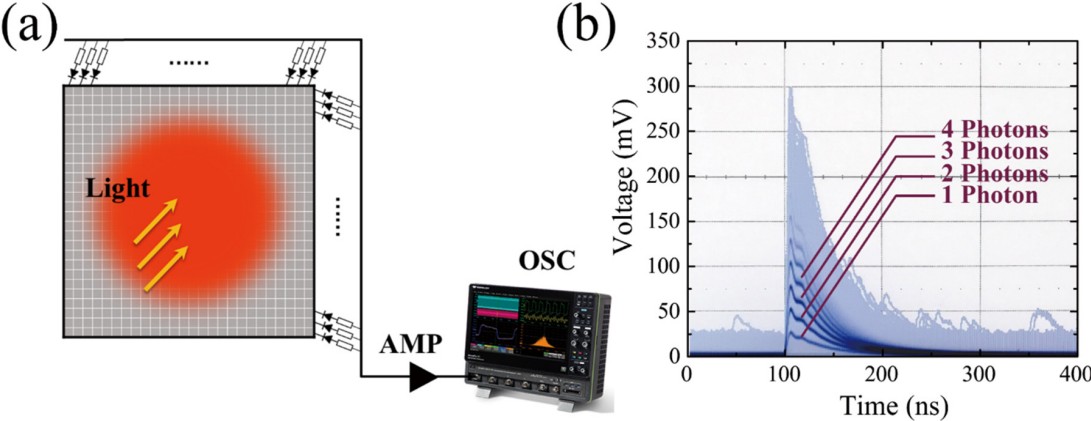

**Figure 1.** (**a**) Structure diagram of MPPC. AMP, amplifier; OSC, oscilloscope. (**b**) Output waveform from MPPC captured by an oscilloscope. The effective photosensitive area of MPPC (C13366-1350GA Hamamatsu) is $1.3 \times 1.3$ mm$^2$ with 667 pixels. (**b**) Output waveform displayed by an oscilloscope. The output pulse duration is about 40 ns.

## 2.2. Quantum Detector Tomography (QDT)

To characterize MPPC detector, we need to determine its corresponding POVM. Different from the partial calibration of PNRDs or the establishment of a complex model based on multiple assumptions, the POVM elements can be directly obtained and applied to describe quantum detectors under very few assumptions by means of QDT [35]. If a series of quantum states $\{\rho\}$ put into the under-test quantum detector is already known, $P_j$ is the probability of obtaining $j$ clicks. According to the input quantum state $\rho$ and the output $j$, the probability $P_j$ can be calculated by:

$$P_j = Tr\left[\Pi^j \rho\right], \tag{1}$$

where a set of semipositive Hermitian operators $\left\{\Pi^j\right\}$ are the POVM elements of the detector, and *Tr* corresponds to the trace. Equation (1) connects the quantum characteristics of the input signal with the classical output of the detector. By measuring a set of known probe states, we are able to characterize an unknown detector and find its POVM elements. POVM elements are complete and non-negative as:

$$\sum_j \Pi^j = 1, \Pi^j \geq 0. \tag{2}$$

In the experiment, coherent state $|\alpha\rangle$ obtained from an attenuated laser is chosen as the alternative to photon number state $|n\rangle$ to be the input state, since the pure photon number state $|n\rangle$ is difficult to prepare. The relationship between the distribution of the coherent state $|\alpha\rangle$ and the photon number state $|n\rangle$ is in accordance with the Poisson distribution:

$$|\alpha\rangle = \sum_0^\infty e^{-\frac{|\alpha|^2}{2}} \cdot \frac{\alpha^n}{\sqrt{n!}} |n\rangle. \tag{3}$$

Different average incident photon number of coherent states can be obtained by varying the attenuation. The range of input quantum states shall include photon number states that saturate the output of the detector so that the POVM space of the detector can be fully covered.

Elements except for the diagonal ones of POVM elements can be assumed as zero since MPPC is phase independent. The equation of the POVM elements is:

$$\Pi^j = \sum_{n=0}^M \theta_n^j |n\rangle\langle n|, \tag{4}$$

where $M$ is the number of photons when the detector is in the saturation state, and $\theta_n^j$ is the diagonal element of POVM elements $\{\Pi^j\}$. Therefore, the probability that the number of incident photons is $i$ and that the number of photons detected by the detector is $j$ can be calculated as:

$$
\begin{aligned}
P_{i,j} &= Tr\left[\Pi^j \rho_i\right] \\
&= Tr\left[e^{-|\alpha_i|^2} \sum_{n=0}^{M} \frac{\alpha_i^n \alpha_i^{*n}}{n!} |n\rangle\langle n| \cdot \sum_{m=0}^{M} \Pi_m^j |m\rangle\langle m|\right] \\
&= Tr\left[e^{-|\alpha_i|^2} \sum_{n=0}^{M} \frac{|\alpha_i|^{2n}}{n!} \Pi_n^j |n\rangle\langle n|\right] \\
&= e^{-|\alpha_i|^2} \sum_{n=0}^{M} \frac{|\alpha_i|^{2n}}{n!} \Pi_n^j.
\end{aligned}
\tag{5}
$$

Here, if $A_{i,n}$ is the density matrix of the input photon number states and $\Pi_{n,j}$ are the POVM elements of MPPC to be measured, then:

$$
A_{i,n} = e^{-|\alpha_i|^2} \frac{|\alpha_i|^{2n}}{n!},
\tag{6}
$$

$$
\Pi_{n,j} = \Pi_n^j,
\tag{7}
$$

and the matrix of probability can be defined as:

$$
P_{i,j} = A_{i,n} \cdot \Pi_{n,j}.
\tag{8}
$$

According to the already known $A_{i,n}$ and the measurement of the corresponding $P_{i,j}$, POVM elements of the detector can be derived from the utilization of the maximum-likelihood estimation (MLE) [39].

There are only two main preconditions for QDT. On the one hand, we need to assume that the detector under test has no memory function. Any previous measurement will not affect the next measurement result, that is, the POVM elements of the detector will not be changed by the measurement, guaranteeing the accuracy of QDT. In terms of MPPC, the time interval between two consecutive measurements of the detector should be greater than the dead time of the detector. On the other hand, we need to assume that truncation of Hilbert Space with infinite dimensions is irrelevant. Selection of the input quantum states should ensure that the POVM space of the detector is completely covered. In terms of MPPC, the maximum photon number of the state space we select makes the output of MPPC saturated. Prior constraints ensure the authenticity of experimental results and the consistency of physical principles.

*2.3. Experimental Setup*

To investigate the performance of MPPC in quantum detection, we did QDT as shown in Figure 2. The pulsed light source is from a mode-locked Ti: sapphire laser at 791 nm. The repetition frequency R is 2 MHz and the pulse width is 2 ps. The other light source at 523 nm with the same pulse width, 2 ps, is obtained by sum-frequency mixing of the Ti: sapphire laser at 791 nm and a pump laser at 1550 nm. The choice of light source is determined in accordance with the parameters of MPPC. The spectral response range of MPPC is from 320 to 900 nm. The pulse repetition frequency is within the range of MPPC's cut-off frequency, and the pulse width is far less than the detector's response time window. An attenuated pulsed laser with intensity at few-photon level was experimentally used to generate the coherent state. The average photon number per pulse could be controlled by a variable attenuator. The powermeter was used to monitor the output power of the laser when the flip mirror is on. The bandpass filter was inserted to remove the background noise photons. The few-photon level light was focused on the photosensitive area of MPPC by a focusing lens f = 50 mm.

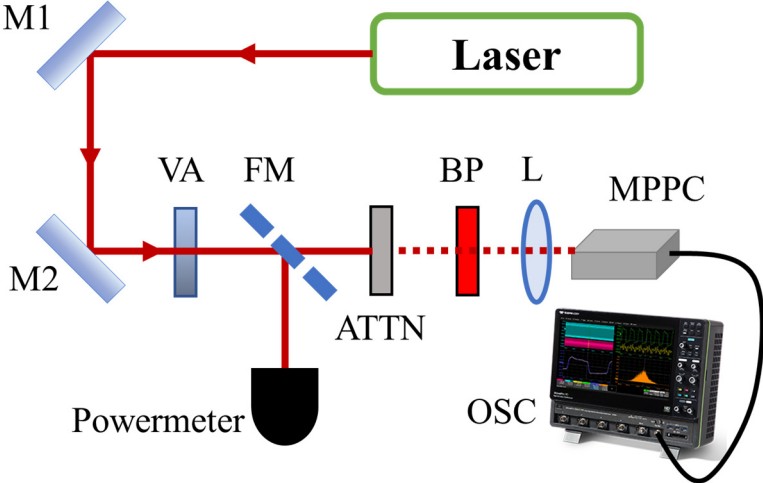

**Figure 2.** Schematic diagram of the experimental setup for MPPC calibration experiment. M1,2: silver mirrors; VA: variable attenuator; FM: flip mirror; ATTN: attenuator with fixed attenuation; BP: band-pass filter; L: focusing lenses (*f* = 50 mm); MPPC: multi-pixel photon counter; OSC: oscilloscope. The pulsed light source is the mode-locked laser at either 791 nm or 523 nm.

The spot radius with two different wavelength incidents on the photosensitive area is approximately 0.64 mm, covering about 500 pixels. After calibrating the flip mirror, the attenuator and the lens, the total transmittance $\gamma$ can be calculated. The photon number reaching MPPC's photosensitive area conforms to $|\alpha|^2 = \frac{\gamma P \lambda}{Rhc}$, in which P is the power of the incident light. We treated the laser pulse as the trigger and recorded the peak amplitude of the output pulse. The probability $P_{n,\alpha}$ can be calculated with the statistical count of different coherent states.

## 3. Results

With the experimental setup mentioned above, MPPC was calibrated at the wavelengths of 791 nm and 523 nm, respectively. The quantum properties of MPPC are demonstrated as follows.

### 3.1. Photon Number Distribution Histogram

Figure 3a,b demonstrate the statistics of MPPC response at the wavelength of 791 nm when the incident average photon number per pulse was about 5.40 and 54.01, respectively. For each averaged incident photon number, we collected 50,000 pulses and recorded the peak amplitude of each pulse in order to construct the histogram. The bin width in statistics is 0.002 V. The peak amplitude histogram of coherent states with different photon number verifies high sensitivity of MPPC and its photon-number-resolving capability. Bounded by the minimum of the voltage distribution, counts of the same peak are added up. The photon number distribution is in accordance with Poisson distribution shown in the insets when the incident photons are relatively low. The red curve is fitted to the experimental data in accordance with a Poisson distribution.

The probability of detecting *i* -photons per pulse is governed by the Poisson distribution:

$$y = \frac{e^j j^i}{i!},\tag{9}$$

where *j* is detected average photon number per pulse. By fitting the histograms in the insets of Figure 3a,b, we obtained *j* = 0.33 photons per pulse when the incident photon flux was 5.40 photons per pulse, and *j* = 3.59 photons per pulse when the incident photon flux was 54.01 photons per pulse, respectively. The *j* divided by the averaged incident photons is the detection efficiency of the detector. The detection efficiency could be roughly deduced to about 6 % at the wavelength of 791 nm. The detection efficiency of the detector at 523 nm was fitted to be about 25 % in the same way.

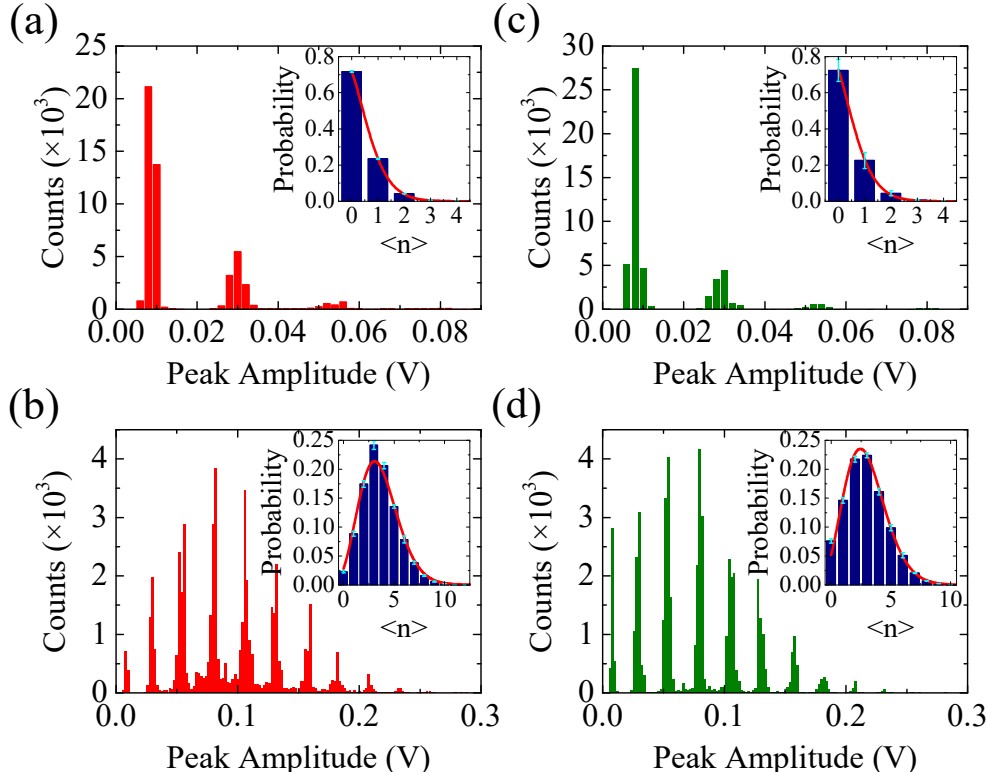

**Figure 3.** Detector voltage amplitude histogram at 791 nm (**a**,**b**) and 523 nm (**c**,**d**). The inset shows the corresponding photon number distribution histogram. Error bars in the insets were obtained by the standard divisions on five independent measurements.

At different incident average photon number, the coefficient of determination is different. The adjusted coefficient of determination $\overline{R}^2$ is used to evaluate the fitting quality, it is computed as $\overline{R}^2 = 1 - \frac{RSS/df_{Error}}{TSS/df_{Total}}$, in which $RSS$ and $TSS$ are the residual sum of squares and total sum of squares, respectively. The closer the value of $\overline{R}^2$ is to 1, the higher the coefficient of determination is. When the incident photons are relatively low, $\overline{R}^2$ is very close to 1, but at large average incident photon number, the coefficient of determination will decrease. We obtain $\overline{R}^2$ is about 0.999 and 0.989 for the incident average photon number per pulse of 5.40 and 54.01, respectively. Therefore, the Poisson fitting could not describe the performance of MPPC completely.

We repeated the experiment at 523 nm by switching the light source to the sum-frequency mixing output beam. As shown in Figure 3c,d, the detector voltage amplitude histogram of MPPC response at the wavelength of 523 nm when the incident average photon number per pulse is about 1.28 and 12.84, respectively. Similarly, we could obtain $\overline{R}^2$ was about 0.999 and 0.993, respectively. The $\overline{R}^2$ value decreased at larger average photon number again.

As the average number of incident photon per pulse increases, the possibility that two photons impinging on the same MPPC pixel will increase. As a result, photon-number-resolving capability of MPPC will decrease. In addition, the cross talk will also influence the photon number distribution detected by MPPC [40]. The detected photon number distribution doesn't obey the Poisson distribution anymore. Therefore, the detection efficiency could not represent very well the performance of MPPC any more at larger detected average photon number per pulse.

### 3.2. POVM

To fully characterize MPPC, QDT was employed to obtain the POVM. In previous theoretical models [36,40,41], the loss, the cross talk and the dark counts of the quantum detector have been

considered separately. However, in the real applications, all these factors affect together on the performance of the detector. By contrast, using QDT, we do not need to consider any information of the under-test detector [38]. All these properties such as the dark counts, the after pulsing, the crosstalk and so forth are involved in QDT process [42]. The reconstruction of POVM has already represented directly the overall relationship between the real known input state and the real detector's output, and this relationship has included the effect of all non-ideal factors. Furthermore, QDT is based on the method called MLE, where no prior assumptions are required [39]. The core of MLE was a kind of iterative algorithm that satisfies Equation (2) as the constraints. The incident light passed through a set of filters served as input quantum state at two different wavelengths. Incident photons ranging from 0.5 to 250 per pulse were collected by adjusting the filters. The digital oscilloscope was connected to the output of MPPC to acquire the information of the output pulses. For simplicity, we chose 0, 1, 2 and ≥3 photons to analyze. Dividing the counts of different photons in the detector voltage amplitude histogram mentioned above by the total number of pulses, we could get the probability of detected different photons and the experimental probability distribution P. In theory, POVM elements of MPPC can be directly deduced with inversion formula of Equation (8). However, in practice, the inverse operation of formula takes a large amount of computation and may not satisfy Equation (2) as the constraints. We set the POVM matrix as a diagonal matrix since MPPC is phase-independent. The initial value of the diagonal element was the reciprocal of clicks number $j$. The zero matrix was used to initialize the photon-number distribution matrix of probe coherent states. Then the initial probability distribution P' was obtained according to Equations (6) and (8). We calculated the square sum of the difference between the experimental probability distribution P and the corresponding initial value P' of each element and kept iterating to make the difference between P and P' close to zero. The reconstructed POVM elements are represented in Figure 4.

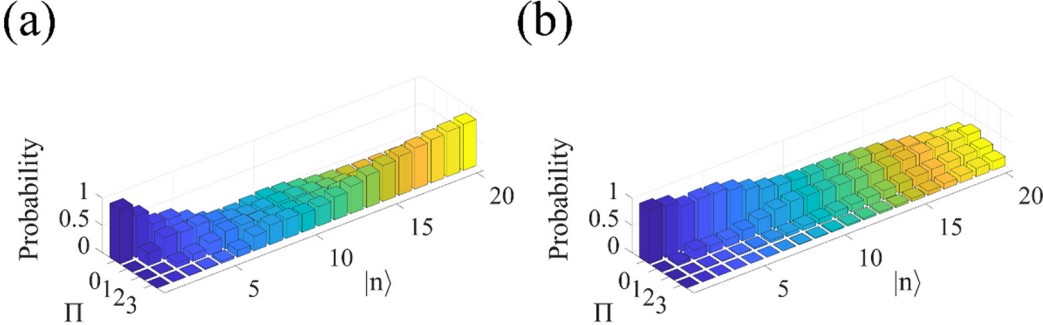

**Figure 4.** POVM elements of MPPC at (**a**) 791 nm; (**b**) 523 nm. The POVM are presented in three dimensions and truncated at twenty for clarity.

According to the reconstructed POVM elements and given quantum states of a input state, the probability distribution can be directly obtained. At different wavelengths, reconstructed POVM elements were used to calculate the output results of MPPC under different average number of incident photons. The calculated results were compared and analyzed with the output results obtained in the experiment, to verify the feasibility and superiority of using QDT method to calibrate spatial multiplexing detectors.

As shown in Figure 5, discrete points represent the probability distribution obtained from the experiment, while solid lines represent the probability distribution calculated by the reconstructed POVM elements. The coincidence degree between experimental results and calculation results is high. The fidelity can be used to quantitatively express the degree of coincidence between the reconstructed POVM elements and the actual quantum characteristics of the detector [38]:

$$F = \sum_{n=0}^{M} \sqrt{P_{exp} \times P_{re}}. \tag{10}$$

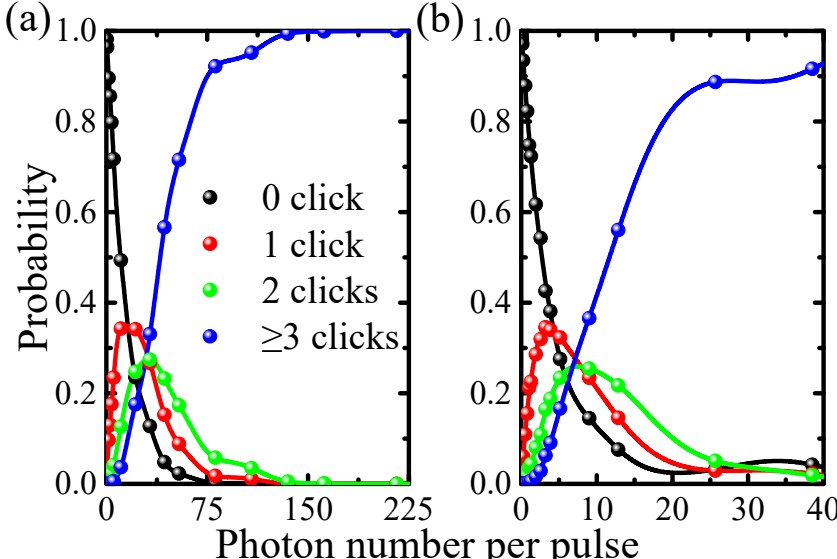

**Figure 5.** Probability distribution of incident photons at (**a**) 791 nm; (**b**) 523 nm. Black, red, green and blue discrete points respectively represent the probability distribution of 0, 1, 2 and ≥3 photons detected by MPPC in the experiment. The corresponding solid lines represent the probability distribution calculated according to the reconstructed POVM elements.

The parameter $P_{exp}$ represents the probability of the *n* output result measured by MPPC in the experiment, while $P_{re}$ represents the probability of the corresponding output result obtained by the reconstructed POVM elements under the same average number of incident photons. The calculated results show that the values of fidelity *F* are all larger than 99.96 % in the interval of 0.5~250 incident photons. Thus, the reconstructed POVM elements accurately characterizes the MPPC. According to Figure 5, the 1-click peak was achieved at 15.9 incident photons per pulse at 791 nm in Figure 5a while 3.6 incident photons per pulse at 523 nm in Figure 5b, showing the difference on the detection efficiency. Therefore, the detection efficiency of MPPC at 523 nm is significantly higher than that at 791 nm. The uncertainties of the reconstructed POVM elements and the probability distributions of incident photons are mainly caused by the laser source and the powermeter to measure the laser power before attenuation. Moreover, the intensities stability of the laser pulse would also destroy the Poisson distribution of the incident photon pulses.

To visualize the quantum performance of the MPPC, we plotted cross section of the Wigner function based on reconstructed POVM [43] in Figure 6. At 523 nm, for one click POVM, the origin of the Wigner function is negative (-0.032), which indicates that MPPC is a fundamental quantum detector. However, for 791 nm, Wigner function is positive at the origin, which means that the quantum characteristics of MPPC is not as obvious as that of MPPC at 523 nm. We ascribe this situation to noise and low detection efficiency of MPPC at this wavelength. For more click POVMs, the origin of the Wigner function is flattened, which is caused by fast decoherence [44].

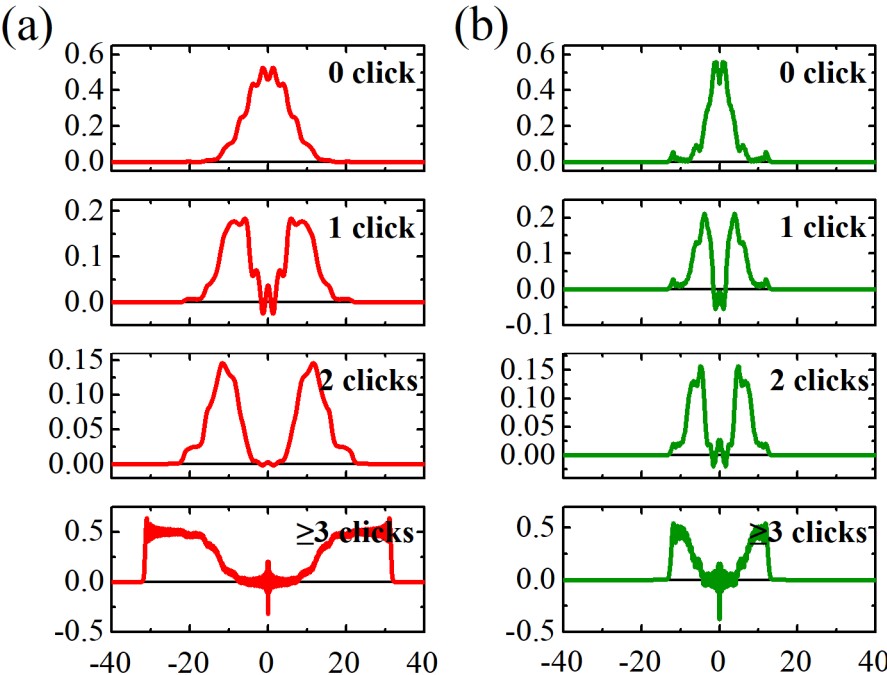

**Figure 6.** Cross section of the Wigner function for MPPC for 0 to ≥3 clicks outcomes at (**a**) 791 nm (red); (**b**) 523 nm (green). The solid lines provide the experimental Wigner function.

## 4. Discussion and Conclusions

We experimentally chose two certain wavelengths of light in the near infrared and the visible bands to calibrate MPPC respectively. On the premise of positional optimization, the initial values of the probability distribution can be reliably measured. Since the initial values have already contained information about unexpected parameters, QDT could calibrate the detector more accurately and completely. In QDT, the reconstruction is completed based on the direct output signals of the detector and MLE. With the method of reconstructing POVM elements, continuous curve of the probability distribution can be predicted from discrete points of experimental data. This kind of calibration maximized the amount of retained information and could be applied to any detector with photon-number-resolving capability, for example in the field of quantum information and metrological applications. Considering different wavelengths, the material of the sensitive surface demonstrates different characteristics. The different quantum efficiencies at the near-infrared and the visible bands result in different signal-to-noise ratios. The decrease of the signal-to-noise ratio leads to the reduction of the photon-number-resolving capability. The larger the average number of photons incident on MPPC is, the more indistinguishable discrete boundaries of the voltage distribution regions are.

In conclusion, we use the method of QDT to calibrate MPPC, which is more accurate and integrated than the previous methods. In the near infrared and visible bands (791 nm and 523 nm), the POVM elements and associated Wigner functions were calculated and faithful quantum characteristics of MPPC were acquired. According to the calibration, the different characteristics of MPPC at different wavelengths are described and discussed. We believe that the quantum calibration of MPPC as a photon-number resolving detector will inform in-depth researches and applications of MPPC and other single photon detectors.

**Author Contributions:** Conceptualization, Y.C. (Yu Chen) and E.W.; methodology, Y.C. (Yu Chen), X.C. and E.W.; software, Y.C. (Yujie Cai), X.C. and J.M.; validation, Y.C. (Yujie Cai); formal analysis, Y.C. (Yujie Cai), Y.C. (Yu Chen), X.C., J.M., G.X. and A.X.; investigation, Y.C. (Yujie Cai), Y.C. (Yu Chen) and Y.W.; resources, E.W.; data curation, Y.C. (Yujie Cai) and Y.C. (Yu Chen); writing—original draft preparation, Y.C. (Yujie Cai) and Y.C. (Yu Chen); writing—review and editing, Y.C. (Yujie Cai), Y.C. (Yu Chen), and E.W.; visualization, Y.C. (Yujie Cai); supervision, E.W.; project administration, Y.C. (Yu Chen) and E.W.; funding acquisition, E.W.

**Funding:** This work is funded by National Natural Science Foundation of China (11722431, 11674099, 11621404); Program of Introducing Talents of Discipline to Universities (B12024); Shanghai International Cooperation Project (16520710600); Natural Science Foundation of Shanghai (16ZR1409400); Shuguang Program (15SG22) by Shanghai Education Development Foundation and Shanghai Municipal Education Commission.

**Acknowledgments:** The authors thank C.D. for the discussion with MPPC settings.

**Conflicts of Interest:** The authors declare no conflict of interest.

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
