# Peer review of "Quantum Calibration of Photon-Number-Resolving Detectors Based on Multi-Pixel Photon Counters"

_applsci, doi:10.3390/app9132638_

Reviewer 1 Report

The article by Yujie Cai et al. presents a full calibration of a multi-pixel photon counter (MPPC) using the method of quantum detector tomography (QDT). I would like to see a better explanation of the novelty of the work and a few revisions for clarity of language, but overall the results and presentation here would be valuable to researchers in a range of fields. I recommend publication after minor revision. Specific comments are below.

Technical comments:

·         In Lines 62-63, the authors should clarify what is meant by “we” in the statement “we can characterize…”, in relation to the cited References 37 and 38, which are works by other groups. I think a stronger case needs to be made within the article for the novelty of this work relative to these earlier papers. From a brief read of both references, my impression (which may not be accurate) is that Ref 37 describes QDT characterization of a different type of PNR detector (TES). Meanwhile, Ref 38 describes characterization of MPPC detector from Hamamatsu (similar to this work) and includes some QDT analysis, but perhaps does not fully calibrate the MPPC using QDT? Presumably there are also differences in the Hamamatsu model that was available 10 years ago. In any case, I recommend the authors clarify which aspects of these References they are intending to cite, and also that they explain and emphasize more strongly the novel aspects of their current work.

·         In Lines 116-117, have the indices been switched in the text?  I think Eq. 5 corresponds to number of incident photons i and number of detected photons j.

Minor readability edits:

Here are some optional suggestions to improve readability and/or clarity of the text.

·         Line 17: change “reliable to” to “reliable to calibrate

·         Line 28: change “generator” to “generation”

·         Line 50: replace “And it” with “MPPC”

·         Line 60: replace “They” with “Lundeen et al.

·         Line 99: replace “It” with “Eq. 1”

·         Line 100: remove “factor” (?)

·         Line 169: change “Voltage” to “Detector voltage” or “MPPC voltage” in the figure caption

·         Line 203: change “was served” to “served”

·         Line 258: this is a bit confusing as written.  Consider replacing “… capability.  Such retained information makes sense in the field of …”  with  “… capability, for example in the field of …

·         Line 260: change “characters” to “characteristics”

·         Line 262: change “The larger average” to “The larger the average”

·         Line 270: consider changing “is consultative for” to “will inform” or “is relevant for

Author Response

Dear Editor,

Thank you very much for your letter and advice. We have corrected the style of our manuscript carefully as you mentioned. We also appreciate the reviewers for their suggestions. We revised our manuscript according to the reviewers’ comments (reviewer 1). Here are the answers to the reviewer’ questions and the modifications in the manuscript.

Reviewer 1

In Lines 62-63, the authors should clarify what is meant by “we” in the statement “we can characterize…”, in relation to the cited References 37 and 38, which are works by other groups. I think a stronger case needs to be made within the article for the novelty of this work relative to these earlier papers. From a brief read of both references, my impression (which may not be accurate) is that Ref 37 describes QDT characterization of a different type of PNR detector (TES). Meanwhile, Ref 38 describes characterization of MPPC detector from Hamamatsu (similar to this work) and includes some QDT analysis, but perhaps does not fully calibrate the MPPC using QDT? Presumably there are also differences in the Hamamatsu model that was available 10 years ago. In any case, I recommend the authors clarify which aspects of these References they are intending to cite, and also that they explain and emphasize more strongly the novel aspects of their current work.

Reply: We emphasis the novelty and importance of this work in the introduction part and clarified the difference between our work and Ref.37 and Ref.38 in the manuscript. We re-ordered the references according to their appearance.

“In previous work [37], the quantum state tomography (QST) has been demonstrated with MPPC, in which the quantum property of the detector was considered as a known fact. The quantum state of the incident photons was reconstructed according to the QST. But in QDT, the quantum property of the detector is unknown and the incident photon states are well-known. The photon-number resolving capability was well characterized by QDT with time-multiplexed superconducting nanowire single photon detector at 1550 nm [36]. Comparing with the previous model description methods [40], we can characterize the quantum properties of MPPC based on spatial-multiplexing technique precisely and comprehensively through the reconstructed POVM elements at two different wavelengths. It will facilitate the application of precisely calibrated MPPC in the field of quantum information.”

36. Natarajan, C.M.; Zhang, L.; Coldenstrodt-Ronge, H.; Donati, G.; Dorenbos, S.N.; Zwiller, V.; Walmsley, I.A.; Hadfield, R.H. Quantum detector tomography of a time-multiplexed superconducting nanowire single-photon detector at telecom wavelengths. Opt. Express 2013, 21, 893-902, doi:10.1364/OE.21.000893.

37. Afek, I.; Natan, A.; Ambar, O.; Silberberg, Y. Quantum state measurements using multipixel photon detectors. Phys. Rev. A 2009, 79, 043830, doi:10.1103/PhysRevA.79.043830.

40. Chen, X.; Ding, C.; Pan, H.; Huang, K.; Laurat, J.; Wu, G.; Wu, E. Temporal and spatial multiplexed infrared single-photon counter based on high-speed avalanche photodiode. Sci. Rep. 2017, 7, 44600, doi:10.1038/srep44600.

In Lines 116-117, have the indices been switched in the text?  I think Eq. 5 corresponds to number of incident photons i and number of detected photons j.

Reply: We corrected the indices in the text.

Therefore, the probability that the number of incident photons is i and that the number of photons detected by the detector is j can be calculated as

Here are some optional suggestions to improve readability and/or clarity of the text.

·         Line 17: change “reliable to” to “reliable to calibrate”

·         Line 28: change “generator” to “generation”

·         Line 50: replace “And it” with “MPPC”

·         Line 60: replace “They” with “Lundeen et al.”

·         Line 99: replace “It” with “Eq. 1”

·         Line 100: remove “factor” (?)

·         Line 169: change “Voltage” to “Detector voltage” or “MPPC voltage” in the figure caption

·         Line 203: change “was served” to “served”

·      Line 258: this is a bit confusing as written.  Consider replacing “… capability.  Such retained information makes sense in the field of …”  with  “… capability, for example in the field of …”

·         Line 260: change “characters” to “characteristics”

·         Line 262: change “The larger average” to “The larger the average”

·         Line 270: consider changing “is consultative for” to “will inform” or “is relevant for”

Reply: We modified the sentences in the manuscript according to the reviewer’s suggestions.

Best regards

Sincerely yours,

E Wu

State Key Laboratory of Precision Spectroscopy

East China Normal University

Reviewer 2 Report

In this paper, the Authors reconstruct the positive operator-valued measure of multi pixel photon counters, which are a commercial class of photon-number-resolving detectors by means of quantum detector tomography. The measurements are performed at two different wavelength, namely at 523 and 791 nm. 

In general, the scope of the paper is clear.

However, I have some requirements in order to improve its quality and better clarify some aspects.

First of all, in the description of the experimental setup the Authors should explain which kind of information is extracted from the oscilloscope. Moreover, since the detector output is amplified, the Authors should specify in the text and in the caption of Fig. 1 how long the temporal duration of the output signal is. 

Secondly, in the first part of Section Results, the Authors present the reconstruction of the photon-number statistics of coherent states of light generated at different wavelengths and with different mean values. In this respect, I am surprised that the Authors did not take into account the modifications of the statistical distributions induced by the nonidealities. For instance, even if the value of the cross-talk probability of the employed MPPC detector is small, it cannot be neglected. The Authors should compare their results to the ones presented in some other papers. For instance, they can have a look at Afek et al., Phys. Rev. A 79, 043830 (2009), Ramilli et al., J. Opt. Soc. Am. B 27, 852 (2010), and at the more recent Chesi et al., Sci. Rep. 9, 7433 (2019), in which the same generation of MPPC is used. Moreover, they should better clarify in which way they deduce the quantum efficiency of the detector at the two investigated wavelengths. 

In the final part of Section Results, the Authors claim that at increasing the number of incident photons per pulse, the detected photon-number distribution does not obey the Poisson distribution anymore. However, I am convinced that this discrepancy cannot be ascribed to pile-up effects for the considered mean values. On the contrary, the Authors should consider the effect of cross talk in their model and check if it can improve their reconstructions. 

Concerning the reconstruction of the POVM elements, did the Authors try to retrieve the Wigner function of the detector outcomes, as addressed in the paper of Lundeen et al. (ref. [35])? In the present version of the paper, it is not clear to me why they can assess that ``the reconstructed POVM elements reflect the quantum characteristics of the actual detector’’.

I also have some technical comments:

- In Fig. 1(b), the Authors should add the horizontal and vertical axes.

- Concerning Fig. 3, the Authors should add the error bars to the experimental photon-number distribution histograms.

- Can the Authors comment on the results presented in Fig. 5? How do they interpret the probability distributions at the two different wavelengths? 

- Can the Authors give some information about the uncertainties of the reconstructed POVM elements (Fig. 4) and of the probability distributions of incident photons (Fig. 5)? 

Once the Authors have dealt with all my issues, I can recommend the publication of the paper in Applied Sciences. 

Author Response

Dear Editor,

Thank you very much for your letter and advice. We have corrected the style of our manuscript carefully as you mentioned. We also appreciate the reviewers for their suggestions. We revised our manuscript according to the reviewers’ comments (reviewer 2). Here are the answers to the reviewer’ questions and the modifications in the manuscript.

Reviewer 2

First of all, in the description of the experimental setup the Authors should explain which kind of information is extracted from the oscilloscope. Moreover, since the detector output is amplified, the Authors should specify in the text and in the caption of Fig. 1 how long the temporal duration of the output signal is.

Reply: We added more details of the experiment and added axes to the Fig.1(b).

“We used digital oscilloscope to capture the peak voltage amplitude of the signal pulses. The oscilloscope was triggered by the synchronization signal from the laser source. The MPPC’s output pulse duration was about 40 ns.”

This figure can be found in PDF file because the figure cannot be inserted here.

Secondly, in the first part of Section Results, the Authors present the reconstruction of the photon-number statistics of coherent states of light generated at different wavelengths and with different mean values. In this respect, I am surprised that the Authors did not take into account the modifications of the statistical distributions induced by the nonidealities. For instance, even if the value of the cross-talk probability of the employed MPPC detector is small, it cannot be neglected. The Authors should compare their results to the ones presented in some other papers. For instance, they can have a look at Afek et al., Phys. Rev. A 79, 043830 (2009), Ramilli et al., J. Opt. Soc. Am. B 27, 852 (2010), and at the more recent Chesi et al., Sci. Rep. 9, 7433 (2019), in which the same generation of MPPC is used.

Reply: We added the discussion on the crosstalk and other noises on QDT in the manuscript.

 “In previous theoretical models [37]-[39], the loss, the cross talk and the dark counts of the quantum detector have been considered separately. However, in the real applications, all these factors affect together on the performance of the detector. By contrast, using QDT, we do not need to consider any information of the under-test detector [40]. All these properties such as the dark counts, the after pulsing, the crosstalk and so forth are involved in QDT process [41]. The reconstruction of POVM has already represented directly the overall relationship between the real known input state and the real detector’s output, and this relationship has included the effect of all non-ideal factors. Furthermore, QDT is based on the method called maximum likelihood estimation, where no prior assumptions are required [42].”

37. Afek, I.; Natan, A.; Ambar, O.; Silberberg, Y. Quantum state measurements using multipixel photon detectors. Phys. Rev. A 2009, 79, 043830, doi:10.1103/PhysRevA.79.043830.

38. Ramilli, M; Allevi, A ; Chmill, V; Bondani, M; Caccia, M; Andreoni, A. J. Opt. Soc. Am. B 27, 852 (2010),

39. Chesi G, Malinverno L, Allevi A, Santoro, R., Caccia, M., Martemiyanov, A., & Bondani, M. Scientific reports, 2019, 9(1): 7433, doi: 10.1038/s41598-019-43742-1

40. Chen, X.; Ding, C.; Pan, H.; Huang, K.; Laurat, J.; Wu, G.; Wu, E. Temporal and spatial multiplexed infrared single-photon counter based on high-speed avalanche photodiode. Sci. Rep. 2017, 7, 44600, doi:10.1038/srep44600.

41. Feito, A.; Lundeen, J.S.; Coldenstrodt-Ronge, H.; Eisert, J.; Plenio, M.B.; Walmsley, I.A. Measuring measurement: theory and practice. New Journal of Physics 2009, 11, 093038, doi:10.1088/1367-2630/11/9/093038.

42. Fiurasek, J. Maximum-likelihood estimation of quantum measurement. Phys. Rev. A 2001, 64, 024102, doi:10.1103/PhysRevA.64.024102.

Moreover, they should better clarify in which way they deduce the quantum efficiency of the detector at the two investigated wavelengths.

Reply: We deduce the quantum efficiency of the detector in the following:

“The probability of detecting i-photons per pulse is governed by the Poisson distribution

This equation can be found in PDF file because the equation cannot be inserted here.

where j is detected average photon number per pulse. By fitting the histograms in the insets of Figure 3(a) and (b), we obtained j = 0.33 photons per pulse when the incident photon flux was 5.40 photons per pulse, and j = 3.59 photons per pulse when the incident photon flux was 54.01 photons per pulse, respectively. The j divided by the averaged incident photons is the detection efficiency of the detector. The detection efficiency could be roughly deduced to about 6 % at the wavelength of 791 nm. The detection efficiency of the detector at 523 nm was fitted to be about 25 % in the same way.”

In the final part of Section Results, the Authors claim that at increasing the number of incident photons per pulse, the detected photon-number distribution does not obey the Poisson distribution anymore. However, I am convinced that this discrepancy cannot be ascribed to pile-up effects for the considered mean values. On the contrary, the Authors should consider the effect of cross talk in their model and check if it can improve their reconstructions.

Reply:

In section 3.1, we added more details about the effect of noise on the photon number distribution.

 “As the average number of incident photon per pulse increases, the possibility that two photons impinging on the same MPPC pixel will increase. As a result, photon-number-resolving capability of MPPC will decrease. In addition, the cross talk will also influence the photon number distribution detected by MPPC [38]. The detected photon number distribution doesn’t obey the Poisson distribution anymore.”

Concerning the reconstruction of the POVM elements, did the Authors try to retrieve the Wigner function of the detector outcomes, as addressed in the paper of Lundeen et al. (ref. [35])? In the present version of the paper, it is not clear to me why they can assess that ``the reconstructed POVM elements reflect the quantum characteristics of the actual detector’’.

Reply: We add the Wigner function as Figure 6.

This figure can be found in PDF file.

 “To visualize the performance of the MPPC, we plotted cross section of the Wigner function based on reconstructed POVM [43] in Figure 6. At 523 nm, for one click POVM, the origin of the Wigner function is negative (-0.032), which indicates that MPPC is a fundamental quantum detector. However, for 791 nm, Wigner function is positive at the origin, which means that the quantum characteristics of MPPC is not as obvious as that of MPPC at 523nm. We ascribe this situation to noise and low detection efficiency of MPPC at this wavelength. For more click POVMs, the origin of the Wigner function is flattened, which is caused by fast decoherence [44].”

43. Sridhar, N; Shahrokhshahi, R; Miller, AJ; Calkins, B ; Gerrits, T ; Lita, A ; Nam, SW ; Pfister, O. JOSA B, 2014, 31(10): B34-B40, doi:10.1364/JOSAB.31.000B34.

44. D’Auria, V., Lee, N., Amri, T., Fabre, C. & Laurat, J. Quantum decoherence of single-photon counters. Phys. Rev. Lett. 107, 050504(2011), doi: 10.1103/PhysRevLett.107.050504.

In Fig. 1(b), the Authors should add the horizontal and vertical axes.

Reply: We add the horizontal and vertical axes in Fig.1(b).

Concerning Fig. 3, the Authors should add the error bars to the experimental photon-number distribution histograms.

Reply: We add the error bars to the experimental photon-number distribution histograms according to the standard divisions of five independent measurements.

This figure can be found in PDF file.

Can the Authors comment on the results presented in Fig. 5? How do they interpret the probability distributions at the two different wavelengths?

Reply: We added the discussion on Fig. 5 in the manuscript.

“According to Figure 5, the detection efficiency of MPPC at 523 nm is significantly higher than that at 791 nm. The 1-click peak was achieved at 15.9 incident photons per pulse at 791 nm in Figure 5a while 3.6 incident photons per pulse at 523 nm in Figure 5b, showing the difference on the detection efficiency.”

Can the Authors give some information about the uncertainties of the reconstructed POVM elements (Fig. 4) and of the probability distributions of incident photons (Fig. 5)?

Reply: We added the discussion about the uncertainties of the reconstructed POVM elements and the probability distribution of the incident photons.

“The uncertainties of the reconstructed POVM elements and the probability distributions of incident photons are mainly caused by the laser source and the powermeter to measure the laser power before attenuation. Moreover, the intensities stability of the laser pulse would also destroy the Poisson distribution of the incident photon pulses.”

Best regards

Sincerely yours,

E Wu

State Key Laboratory of Precision Spectroscopy

East China Normal University

Round  2

Reviewer 2 Report

In the revised version of the paper, the authors successfully addressed all the issues raised in my first report. 

According to me, the paper can now be published in Applied Sciences.